# Coevolution of Rumen Epithelial circRNAs with Their Microbiota and Metabolites in Response to Cold-Season Nutritional Stress in Tibetan Sheep

**DOI:** 10.3390/ijms231810488

**Published:** 2022-09-10

**Authors:** Xinyu Guo, Yuzhu Sha, Xiaoning Pu, Ying Xu, Liangwei Yao, Xiu Liu, Yanyu He, Jiang Hu, Jiqing Wang, Shaobin Li, Guoshun Chen

**Affiliations:** Gansu Key Laboratory of Herbivorous Animal Biotechnology, College of Animal Science and Technology, Gansu Agricultural University, Lanzhou 730070, China

**Keywords:** circRNAs, rumen epithelium, rumen microbiota, metabolites, nutrition stress

## Abstract

This study explores the effects of the coevolution of the host genome (the first genome) and gut microbiome (the second genome) on nutrition stress in Tibetan sheep during the cold season. The rumen epithelial tissue of six Tibetan sheep (Oula-type) was collected as experimental samples during the cold and warm seasons and the study lasted for half a year. The cDNA library was constructed and subjected to high-throughput sequencing. The circRNAs with significant differential expression were identified through bioinformatics analysis and functional prediction, and verified by real-time quantitative PCR (qRT-PCR). The results showed that a total of 56 differentially expressed (DE) circRNAs of rumen epithelial tissue were identified using RNA-seq technology, among which 29 were significantly upregulated in the cold season. The circRNA-miRNA regulatory network showed that DE circRNAs promoted the adaptation of Tibetan sheep in the cold season by targeting miR-150 and oar-miR-370-3p. The results of correlation analysis among circRNAs, microbiota, and metabolites showed that the circRNA NC_040275.1:28680890|28683112 had a very significant positive correlation with acetate, propionate, butyrate, and total volatile fatty acid (VFA) (*p* < 0.01), and had a significant positive correlation with Ruminococcus-1 (*p* < 0.05). In addition, circRNA NC_040256.1:78451819|78454934 and metabolites were enriched in the same KEGG pathway biosynthesis of amino acids (ko01230). In conclusion, the host genome and rumen microbiome of Tibetan sheep co-encoded a certain glycoside hydrolase (β-glucosidase) and coevolved efficient VFA transport functions and amino acid anabolic processes; thus, helping Tibetan sheep adapt to nutrient stress in the cold season in high-altitude areas.

## 1. Introduction

Tibetan sheep are an important livestock genetic resource on the Qinghai-Tibet Plateau and the main source of meat, wool, fuel, and economy for local herdsmen, providing important resources for local residents. Tibetan sheep live on the Qinghai-Tibet Plateau at an altitude of more than 2500 m and naturally graze throughout the year. It has a unique ecological environment, with high altitude, low temperature, low oxygen content, and strong ultraviolet radiation, etc. [1]. The grassland pastures in the alpine pastoral areas of the Qinghai-Tibet Plateau experience both a cold season and a warm season, corresponding to the green grass stage (May–October) and withering grass stage (October–April of the following year). The phenological changes in plateau vegetation on the Tibetan Plateau significantly affect grassland ecosystems and the growth and development of grazing livestock [2]. During the green grass stage, the pasture grass is abundant and nutrient-rich, with high protein and carbohydrate contents [3]. However, during the withering period, the main food is withered grass, and the grassland is seriously low in nutrients. Therefore, nutrient stress in the cold season is one of the main plateau ecological factors that limits the development and utilization of Tibetan sheep germplasm resources.

The rumen is an important digestive organ of ruminants that can promote the growth and development of ruminants, improve their production performance, and maintain the health of the host. The various metabolites produced by rumen microbiota can be effectively digested and absorbed by the rumen epithelium; thereby providing energy and nutrients for the body and maintaining the normal digestion and absorption of the body; this is beneficial to Tibetan sheep under nutritional stress during the cold season [4]. The study found that high-altitude mammals have evolved strong absorption and transport of volatile fatty acids (VFAs); thus, revealing a new mechanism of adaptive evolution of the “first genome” (own genome) and “second genome” (gut microbiome) from the perspective of energy metabolism [5]. Previous studies on the genetic mechanism of high-altitude adaptation have mostly focused on the variation in animal genetic material [6], and less attention has been given to the important contributions of the “second genome” that coevolved with hosts, especially in energy metabolism.

CircRNAs, as a class of endogenous noncoding RNA molecules with a closed loop structure, participate in cell proliferation, differentiation, and apoptosis, and regulate the expression of many aspects of gene expression, such as transcription, post-transcription, and epigenetics [7]. CircRNA inhibits the activity of miRNA by acting as a competitive endogenous RNA (ceRNA) or “miRNA sponge”; thereby adsorbing and inhibiting miRNA expression and affecting the molecular level of downstream target genes [8]. One study found that circRNAs could be expressed in intestinal epithelial cells, participate in intestinal homeostasis and immune function [9], and also participate in adipogenesis and thermogenesis [10], and then regulate intestinal epithelial barrier function by miRNA adsorption by sponging [11]. In the maintenance of intestinal epithelial homeostasis, circRNA (circHIPK3), as a major regulator of intestinal epithelial repair after acute injury, could promote intestinal epithelial homeostasis by reducing the function of miRNA 29b [12]. In addition, circRNAs can bind or regulate RNA binding proteins (RBPs) or other proteins to form circRNA-protein complexes, which, in turn, affect cell function by affecting the function and mechanism of binding proteins [13]. CircRNA-Maml2 activated the downstream Wnt/β-catenin pathway, promoted the proliferation and migration of CT26 cells, and then promoted the repair of intestinal mucosa [14]. At present, studies on circRNAs have mainly focused on the gastrointestinal tract of humans [11] and mice [14], while studies related to circRNAs in the rumen epithelium of ruminants have rarely been reported. Therefore, based on previous research [4,15], this study revealed nutritional stress of Tibetan sheep in the cold season through the interaction of rumen epithelial circRNAs with microbiota and metabolites, further explored the coevolutionary effect between the host genome and microbial genome, and provided a basis for analyzing the response of Tibetan sheep to nutritional stress in the cold season on the Qinghai-Tibet Plateau.

## 2. Materials and Methods

### 2.1. Ethics Statement

All animals involved in the study were in accordance with the regulations for the Administration of Affairs Concerning Experimental Animals (Ministry of Science and Technology, Beijing, China; revised in June 2004), and the sample collection protocols were approved by the Livestock Care Committee of Gansu Agricultural University (Approval No. GAU-LC-2020-27).

### 2.2. Experimental Design and Sample Collection

Six healthy Tibetan sheep ewes were randomly selected from the same pasture in Hezuo City, Gannan Tibetan Autonomous Prefecture, Gansu Province, China (3300 m above sea level), which uses local traditional natural grazing management. The selected ewes had the following conditions: similar weight (35.12 ± 1.43 kg), good health, and the same age (1 year old). These animals were randomly divided into two groups, representing the warm (July) and cold (December) seasons, with three sheep in each period, grazing in the same pasture as the other sheep. Samples were collected in July 2019 and December 2019. Before grazing in the morning, the local traditional jugular vein bloodletting method was used. Subsequently, the rumen abdominal sac tissue block (1 cm × 1 cm) was collected, and the rumen contents were quickly removed by rinsing with normal saline. At the same time, after cutting a small piece of rumen abdominal sac, the contents were rinsed with PBS (phosphate-buffered saline), and then the epithelial tissue was separated with blunt scissors. After the samples were processed, they were stored in liquid nitrogen until subsequent extraction of total RNA.

### 2.3. Determination of VFAs, Microbiota Structure, and Differential Metabolites in Rumen

The determination methods and results of microbiota, metabolites, and VFAs are shown in Liu et al. [4]. The two-step library construction method was used to construct and sequence the rumen microbiota [16]. All amplified products were sequenced and analyzed on an Illumina MiSeq platform (Illumina, San Diego, CA, USA). VFAs were determined with a gas chromatograph (GC-2010 plus; Shimadzu, Japan). The internal standard method was adopted, using 2-ethyl butyric acid (2EB) as the internal standard.

### 2.4. RNA Isolation, Library Preparation, and circRNA Sequencing

Illumina technology was used to study the circRNA expression profile and to construct a comprehensive atlas of circRNA expression in the rumen epithelial tissues of Tibetan sheep. In this study, rumen epithelial tissue samples from 6 Tibetan sheep were sequenced. Samples of three qualified RNAs were selected for each group, and a total of 6 cDNA libraries were constructed and coded as cold1, cold2, cold3, warm1, warm2, and warm3. Trizol reagent (Invitrogen, Grand Island, NY, USA) was used to isolate and purify total RNA from each sample. Agilent 2100 Bioanalyzer (Agilent Technologies, Palo Alto, CA, USA) was used to assess RNA quality, which was assessed using ribonuclease-free agarose gel electrophoresis. Subsequently, paired-end sequencing of each sample cDNA library was performed using the Illumina HiSeq2500 platform according to the instructions of BioMarker Technologies (Beijing, China) [17].

### 2.5. Identification of circRNAs

Since splicing sites of circRNA loops cannot be directly compared with the genome, find_circ [18] software first took 20 bp from both ends of each read that could not be compared with the genome as anchor points, and then compared with the genome as independent reads to find the unique matching site. If the alignment positions of the two anchor points were in the opposite direction linearly, the reads of the anchor points were extended until the binding position of the circRNAs was found. If the sequences on both sides at this time were GT/AG splicing signals [19], they were judged as circRNAs.

### 2.6. Identification of Differentially Expressed circRNAs

SRPBM was used as a standardized method to quantify the expression of circRNAs, and DESeq^2^ [20] was used to identify circRNAs with significant differences between cold and warm seasons. Differentially expressed (DE) circRNAs were detected by comparison, and in any pairwise comparison, circRNAs were considered significantly different if *p* <0.05 and fold change >1.5. SRPBM is calculated as follows: SRPBM = (SR × 109)/N. In the above formula, SR refers to the number of splicing reads and N refers to the total number of map reads in these samples.

### 2.7. KEGG and GO Analysis of the Source Genes of DE circRNAs

Kyoto Encyclopedia of Genes and Genomes (KEGG) enrichment analysis and Gene Ontology (GO) enrichment analysis for the source genes of DE circRNAs were used to explore the potential biological functions of circRNAs. ClusterProfiler [21] was used to analyze the biological process, molecular function and cell components of genes from the source genes of circRNAs. Enrichment analysis used hypergeometric tests to find GO entries that were significantly enriched compared to the whole genome background. The application of ClusterProfiler [21] was used to analyze enrichment pathway of DE circRNAs.

### 2.8. CircRNA Targeted miRNA Loci Analysis

The miRNA sites of circRNAs were predicted using targetscan [22] and miranda [23]. A circRNA-miRNA network was established based on sequencing data, as well as previously predicted miRNA binding sites. The circRNA-miRNA network was constructed using Cytoscape software.

### 2.9. RT-qPCR Analysis

Total RNA in this study was extracted from rumen epithelial tissue samples of Tibetan sheep with TRIzol reagent (Invitrogen, Carlsbad, CA, USA), and cDNAs were synthesized using RT-qPCR kit (Takara, Dalian, China). Six primers were designed to amplify circRNA-specific backsplice junctions (Table 1). PCR products were first identified by 1.5% agarose gel electrophoresis, followed by extraction of product bands and Sanger sequencing. Then, the above 6 circRNAs were selected to verify the reliability of RNA-seq by real-time fluorescence quantification (RT-qPCR). RT-qPCR analysis was performed by using 2 × ChamQ SYBR qPCR Master Mix (Vazyme, Nanjing, China) on Applied Biosystems QuantStudio^®^6 Flex (Thermo Lifetech, Waltham, MA, USA), and the relative expression levels of source genes were analyzed by using the 2^−ΔΔCT^ method in triplicate. Internal reference gene for sheep was β-actin.

### 2.10. Statistical Analysis

All statistical analyses were performed using IBM SPSS 22.0 (SPSS, Inc., Chicago, IL, USA). The two-tailed test was used for correlation analysis between circRNA and VFA in SPSS software. The microbiota at the genus level (relative abundance > 0.5%) and DE circRNAs were selected for cluster analysis (correlation threshold < 0.1, *p* < 0.05). The interaction between differential metabolites and DE circRNAs was explored through functional pathway maps of co-enrichment of differential metabolites and source genes of DE circRNAs.

## 3. Results

### 3.1. Identification of Rumen Epithelium circRNAs

A total of 101.41 Gb of clean data was obtained, and the percentage of Q30 bases in each sample was greater than 98.22% (Appendix A). The comparison statistical results showed that the comparison efficiency between reads of each sample and reference genome ranged from 95.05% to 98.84%, and the selected reference genome met the needs of subsequent analyses (Appendix A). After filtering the raw data, a total of 265,595 high-quality clean reads were obtained, and the total number of clean reads in cold and warm seasons was 117,265 and 148,330, respectively. The study identified a total of 34,955 circRNAs, with 15,578 and 19,377 circRNAs in the cold and warm seasons, respectively. Most circRNAs were derived from genes encoding exons, and the rest were derived from antisense strands and introns (Figure 1B). CircRNAs with a length of 400 bp were the most numerous and 93.32% of circRNAs were between 200 and 800 nt in length (Figure 1B). Chromosome distribution results showed that these circRNAs were mostly distributed on 28 chromosomes, mainly in NC-040260.1 on chromosomes (Figure 1C).

### 3.2. Analysis and Validation of DE circRNAs

To explore the regulatory role of circRNAs in the rumen epithelial development of Tibetan sheep, the expression of differentially expressed (DE) circRNAs was analyzed by calculating SRPBM values. A total of 56 DE circRNAs were found in this study, of which, 29 were significantly upregulated and 27 were significantly downregulated in the cold season (Appendix A). The scatter plot (Figure 1D) and cluster heatmap (Figure 1A) visually showed the overall distribution of DE circRNAs.

Six DE circRNAs were randomly selected for RT-qPCR detection, and divergent primers containing their linking sites were synthesized (Figure 2A). The RT-qPCR results showed that the expression profiles of these six circRNAs were consistent with the RNA-Seq results (Figure 2C). Sanger sequencing of the qPCR product fragments showed that the circular junction sequence sites of these circRNAs were completely consistent with the sequencing results of circRNAs (Figure 2B). This indicated that the sequencing data and expression of the circRNAs identified in this study were reliable.

### 3.3. KEGG Pathway and GO Enrichment for Source Genes of DE circRNAs

As shown in Figure 3A, KEGG enrichment analysis showed that the source genes of DE circRNAs were significantly enriched in 14 pathways, mainly in pathways related to amino acid and protein synthesis, such as lysine degradation (ko00310), lysosome (ko04142), biosynthesis of amino acids (ko01230), and protein processing in endoplasmic reticulum (ko04141). As shown in Figure 4, GO enrichment analysis found that a total of 95, 25, and 20 GO terms were significantly enriched in biological processes (Figure 3B), cellular components (Figure 3C), and molecular functions (Figure 3D), respectively. In addition, the most source genes of DE circRNAs were significantly enriched in pathways related to calcium ion regulation in terms of biological processes.

### 3.4. Construction of circRNA-miRNA Regulatory Network

In order to further analyze the biological functions of DE circRNAs, a circRNA-miRNA network was constructed by integrating miRNAs. A total of three upregulated circRNAs and six downregulated circRNAs were selected from 56 DE circRNAs to study the interaction of circRNAs with their targeted miRNAs. As shown in Figure 5, a total of 44 miRNAs were targeted by 9 circRNAs with significant differences, ranging from 1 targeted miRNA in NC_040262.1:53244324|53251250 to 16 targeted miRNAs in NC_040269.1:68309959|68364662. Among them, oar-miR-370-3p was targeted by seven DE circRNAs, oar-miR-134-3p was targeted by four DE circRNAs, and oar-miR-150 was only targeted by circRNA NC_040269.1:68309959|68364662. Several miRNAs previously reported to be associated with fat metabolism and immunity were identified in this analysis, including oar-miR-370-3p and miR-150.

### 3.5. Interaction Analysis of circRNAs in the Rumen Epithelium and VFAs

Combined with our previous results of rumen VFA measurements [4], correlation analysis was conducted between circRNAs of the rumen epithelium and VFAs (Figure 6). The correlation results showed that the upregulated circRNAs NC_040269.1:68309959|68364662, NC_040275.1:28680890|28683112, and NC_040253.1:45701039|45703283 in the cold season were positively correlated with acetate, propionate, butyrate, and total VFAs, respectively (*p* < 0.01). The downregulation of circRNA NC_040264.1:34000774|34006459 in the cold season was significantly negatively correlated with acetate, propionate, butyrate, and total VFAs (*p* < 0.01).

### 3.6. Combined Analysis of the Rumen Epithelium circRNA-Microbiome-Metabolome

To further explore the differences in genetic characteristics, physiological activities, and energy metabolism of Tibetan sheep during altitude acclimatization, this study conducted correlation analysis among circRNAs of rumen epithelium, previously measured rumen microbiota, and rumen microbial metabolites [4,15]. As shown in Figure 7A, there was a correlation between circRNAs and rumen microbiota. Among the ten DE circRNAs randomly selected, four DE circRNAs (NC_040257.1:64017417|64022902, NC_040264.1:34000774|34006459, NC_040276.1:10040393|10059385, NC_040258.1:11318705|11337918) were significantly positively correlated with Butyrivibrio-2, Ruminococcaceae-NK4A214-group and Succiniclasticum (*p* < 0.05), and were significantly negatively correlated with Rikenellaceae-RC9-gut-group and Ruminococcus-1 (*p* < 0.05). The correlation between the remaining six circRNAs and the above microbiota was the opposite. Moreover, this study found that circRNA NC_040275.1:28680890|28683112 had a significant negative correlation with Prevotallaceae-UCG-003 (*p* < 0.05).

As shown in Figure 7B, DE circRNAs were also correlated with rumen microbial metabolites. The source gene *LOC105605805* was enriched in the lysosome (ko04142) pathway, and was involved in the decomposition of bilirubin and biliverdin in the porphyrin and chlorophyll metabolism pathways (ko00860). The source gene *MAN1A2* was significantly upregulated in the cold season, while the differential metabolite PP-Dol was regulated by gene-*MAN1A* and involved in N-glycan degradation. The source genes of circRNAs that were significantly upregulated in the cold season were mainly enriched in the processes of amino acid biosynthesis and lysine biosynthesis. As shown in Figure 7B, gene-*MAT2B* can regulate methionine to generate homocysteine and is enriched in the biosynthesis of amino acids (ko01230), while the upregulated metabolites serine and proline were also enriched in the same pathway in the cold season. Furthermore, the source genes *KMT2C* and *KMT5B* were also found to regulate lysine biosynthesis.

## 4. Discussion

Based on the coevolution of the host genome and microbial genome, this study explored the role of circRNAs in regulating rumen epithelial development, the microbiota and its metabolites, and further revealed the regulatory mechanism of Tibetan sheep adaptation to nutrition stress in the cold season. The rumen epithelium is an important immune barrier for ruminants and the main site for nutrient absorption and metabolism [24]. This study found that 93.32% of the circRNAs identified in the rumen epithelium were derived from coding exons, which made these source genes have a certain cycle preference and specificity [25], and further participated in the regulation of the development and metabolism of the rumen epithelium. This study further found GO enrichment for source genes of DE circRNAs and showed that source genes were significantly enriched in GO pathways related to calcium ion channels; the source gene *CACNB2* encodes the α1 subunits and auxiliary β, α2δ and γ subunits in the calcium channel complex [26]. The rumen is the main site for the active absorption of calcium ions in sheep, and the Ca^2+^/H^+^ exchange mechanism existing in the rumen epithelial cells depends on the absorption of SCFA [27], which further indicates that Tibetan sheep regulate the absorption of nutrients and energy in rumen epithelial cells through the calcium ion pathway; thereby adapting to nutrient stress in the cold season. Calcium channels play a significant role in adaptation to the plateau environment, and convergent evolution among plateau animal populations, such as with *RYR3*, *RYR2*, and *GRIN2B*, has been implicated in the regulation of calcium channels in Tibetan mastiffs, Tibetan chickens, and Tibetan pigs, respectively [28].

KEGG enrichment analysis showed that the source genes of DE circRNAs were significantly enriched in the lysine degradation (KO00310) pathway, which is mainly affected by different enzymes, and the source genes KMT2C and KMT5B encode histone methyl transferases, respectively [29,30]. Histone methyl transferases have important regulatory roles in multiple biological processes, such as DNA repair, cell cycle regulation, and DNA replication through histone lysine methylation (H4K20) [31]. Gene-KMT2C can be used as a biomarker for predicting disease and plays an important role in disease treatment [32]. In this study, Tibetan sheep participated in immune regulation through lysine degradation pathway in order to adapt to the cold season environment. In addition, the source gene-RFC1 of circRNA NC_040257.1:64017417|64022902 enriched in KEGG pathway Mismatch repair (ko03430) is a typical folate carrier and mediates antifolate transport [33,34], and can maintain mammalian cells’ normal metabolism, energy, differentiation, and growth status [35,36]. The content of folic acid is affected by the ratio of feed to concentrate [37]; the downregulation of circRNA NC_040257.1:64017417|64022902 in the cold season is related to nutritional stress in the cold season.

Source genes of DE circRNA affect nutritional stress in the cold season of Tibetan sheep through immune regulation; gene-*MAP9* and gene-*EVI5* can participate in the synthesis of tubulin, which is a highly conserved component of microtubules and can regulate microtubule dynamics and stability [38,39,40]. Furthermore, microtubules are involved in cell division, formation, movement, and intracellular transport [40]. Gene-*RC3H2* participates in the synthesis of RNA-binding protein-Roquin, binds to immune-related miRNA targets, regulates its stability, and affects the function of macrophages and the peripheral homeostasis of T and B cells [41], indicating that Tibetan sheep can cope with nutritional stress in the cold season by improving the immune response ability of the rumen epithelium; thereby improving the adaptive ability of the Tibetan sheep in the cold season. In addition, circRNAs also play an important role in transcriptional regulation through the ceRNA mechanism [42]. Ding et al. found that circ_0010283 regulated the viability and migration of vascular smooth muscle cells induced by oxidized low-density lipoprotein by targeting miR-370-3p [43]. Ren et al. found that circ_0023461 protected cardiomyocytes from hypoxia-induced dysfunction by targeting miR-370-3p [44]. In this study, DE circRNA NC_040269.1:68309959|68364662 could target miR-150 and miR-370-3p, Oar-miR-370-3p inhibited fat accumulation, which might result from the inhibition of saturated fatty acids that promoted the accumulation of polyunsaturated fatty acids [45]. miR-150 regulated lipid metabolism and the inflammatory response, and also regulated cellular metabolic activity and inflammatory response in animals [46]. Therefore, DE circRNAs of rumen epithelial can improve the lipid metabolism and immune defense ability of Tibetan sheep in the cold season by combining targeting miR-150 and oar-miR-370-3p, indicating that Tibetan sheep can actively respond to nutritional stress in the cold season.

Hosts not only contain a large diversity of microorganisms, but also coevolved with them in response to the external environment [47,48]. Moreover, adaptive evolution affects genes related to energy metabolism and helps ruminants survive at high altitudes [5,6]. Therefore, to further reveal the coevolution between the host genome and the microbial genome, this study analyzed the interaction between circRNAs of the rumen epithelium and microbiota. The source gene, *LOC105605805*, encodes a beta-glucuronidase-like enzyme that catalyzes the gradual degradation of complex carbohydrates and glycoconjugates in mammals to provide the host with nutrients and energy [49]. Meanwhile, Gloux et al. found that the genomes of Ruminococcaceae, Lachnospiraceae, and Clostridiaceae could also encode β-glucosidase in rumen microorganisms [50]. The glycoside hydrolase family plays a key role in the process of microbial colonization, and the symbiotic flora of the intestinal epithelium collaborates with other intestinal symbionts by secreting a wide range of glycosidases and then forming biofilms and colonizing the intestine [51], and at least 81 distinct families of glycoside hydrolases were found in the human distal gut microbial metabilome [52]. This further showed that the host genome and rumen microbial genome of Tibetan sheep were beneficial to microbial colonization by co-coding glycoside hydrolase to cope with harsh environments in the cold season. Furthermore, an important feature of host–microbe coevolution was metabolic collaboration, where the host and its microorganisms depended on each other to be involved in the amino acid and vitamin biosynthesis pathway [48]. The rumen microbiota coevolved with the host genome by expanding gene families related to energy metabolism, such as protein metabolism or carbohydrate metabolism [5]. Zhang et al. found that during the adaptation evolution of plateau animals, VFA production pathways were significantly enriched, and transcriptome analysis showed that genes related to VFA transport were significantly upregulated, and further found that high-altitude ruminants have evolved efficient VFA transport functions [5]. This is consistent with the results of this study, which showed that Tibetan sheep also evolved efficient VFA transport functions to cope with nutrient stress in the cold season. In addition, the interaction results between circRNAs and metabolome showed that *MAT2B* and the differential metabolites serine and proline were enriched in the same KEGG pathway biosynthesis of amino acids (ko01230) and could regulate the production of homocysteine. Zhao et al. found that *MAT2B* promoted adipogenesis by regulating the level of S-adenosylmethionine (SAMe) in cells [53]. SAMe produced the final product, homocysteine, by participating in methylation [54]. Zhou et al. found that the main mechanism by which serine supplementation relieves oxidative stress was to synthesize cysteine through condensation with homocysteine and provided a carbon unit for homocysteine methylation [55]. The DE circRNA NC_040256.1:78451819|78454934 was significantly upregulated in the cold season, and the differential metabolites serine and proline were also significantly upregulated in the cold season, which indicated that Tibetan sheep have evolved efficient amino acid synthesis and metabolism in the cold season. Therefore, as shown in Figure 8, this study further explored the coevolution between the host genome and the rumen microbiome through the connection between circRNAs and microbial metabolites, and this interaction can help Tibetan sheep cope with the harsh environment in the cold season and provide a basis for research on the nutritional stress of Tibetan sheep in the cold season.

## 5. Conclusions

Under the conditions of nutrient stress during the cold season on the Qinghai-Tibetan Plateau, circRNAs in the rumen epithelium and rumen microbiota and their metabolites coped with nutrient stress through coevolution. The source genes of DE circRNAs in the rumen epithelium improved the energy metabolism and immune defense of Tibetan sheep in the cold season, and then affected the cold season nutritional stress of Tibetan sheep by enriching the KEGG pathways, such as lysine degradation and GO pathways related to calcium ion channels and targeting miR-150 and oar-miR-370-3p. The interaction of rumen epithelial circRNAs with rumen microorganisms and their metabolites indicated that the Tibetan sheep host genome and rumen microbiome co-encoded a certain glycoside hydrolase (β-glucosidase) to facilitate the colonization of microorganisms and to cope with the harsh environmental conditions in the cold season. In addition, the Tibetan sheep host genome and rumen microbiome have coevolved efficient VFA transport functions and amino acid synthesis and metabolism processes; thereby helping Tibetan sheep adapt to the nutrient stress in the cold season in high-altitude areas. Therefore, this study aimed to explore the adaptation and regulation of Tibetan sheep to nutrient stress in the cold season by analyzing the interaction between the host genome and the gut microbiome and to provide a basis for research on the adaptation of ruminants in the cold season in high-altitude areas.

## Figures and Tables

**Figure 1 ijms-23-10488-f001:**
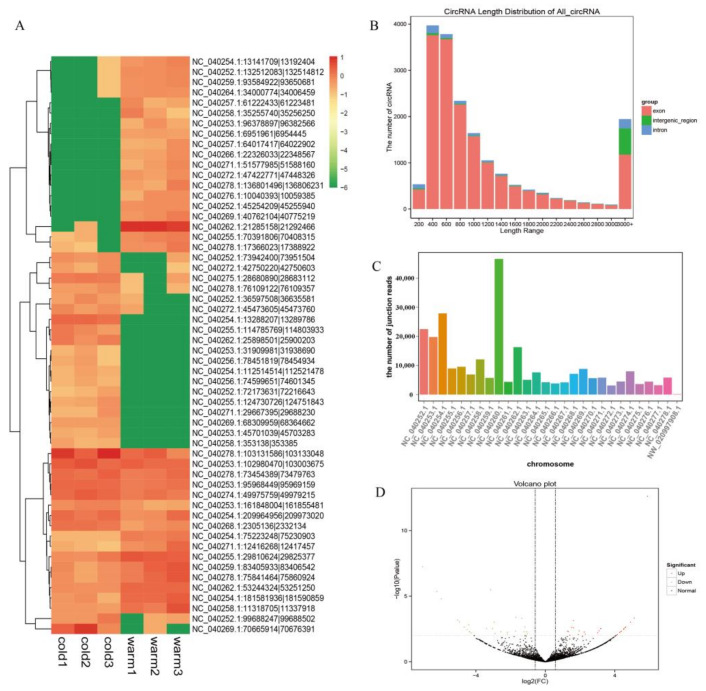
Summary of circRNAs analysis. (**A**) Cluster heat map of DE circRNAs expression. Red means upregulation, green means downregulation; (**B**) the length distribution of circRNAs; (**C**) statistics of circRNAs sources. *X*-axis represents chromosome and the *Y*-axis represents the number of circular RNA junction reads on the corresponding chromosome; (**D**) volcano plot of all DE circRNAs between the cold and warm groups. *X*-axis represents the value of log2 (FC) and the *Y*-axis represents the value of −log10 (*p*-value).

**Figure 2 ijms-23-10488-f002:**
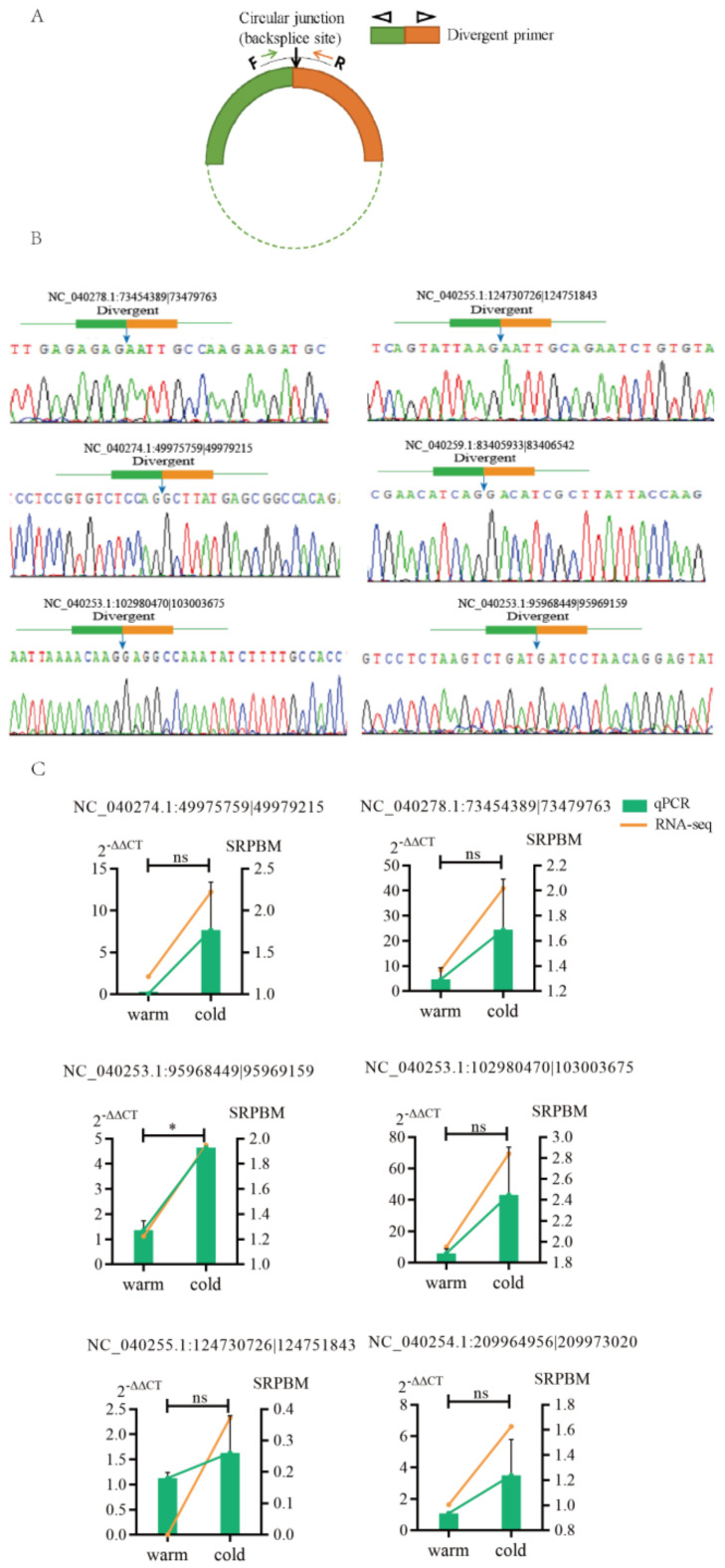
Verification of randomly selected six circRNAs from RNA–Seq. (**A**) The circular junctions were amplified using divergent primers; (**B**) circular junctions were confirmed by Sanger sequencing of RT–PCR products using divergent primers; (**C**) the level of expression of circRNAs from RNA–Seq compared with those obtained by RT–qPCR. RT–qPCR data are presented as mean ± S.D. Asterisks (*) and ns (non-significant) indicate *p* < 0.05 and *p* > 0.05, respectively.

**Figure 3 ijms-23-10488-f003:**
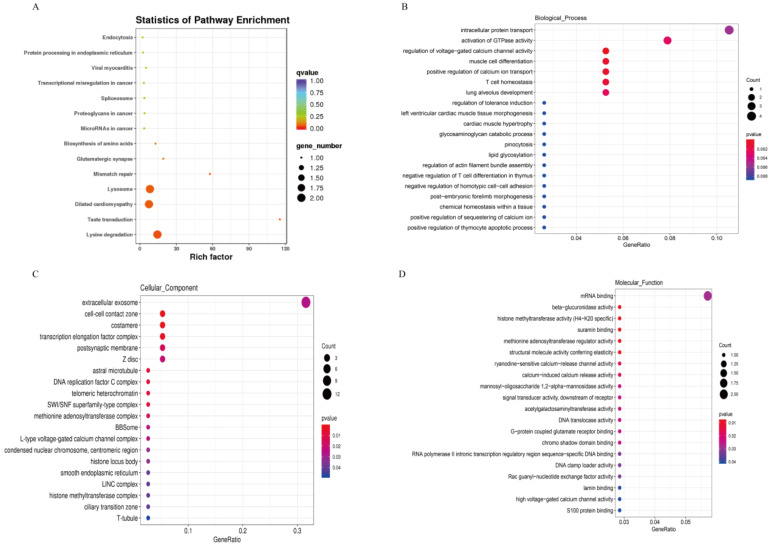
(**A**) KEGG enrichment analysis for source genes; (**B**) GO terms in biological processes; (**C**) GO terms in cellular components; (**D**) GO terms in molecular functions.

**Figure 4 ijms-23-10488-f004:**
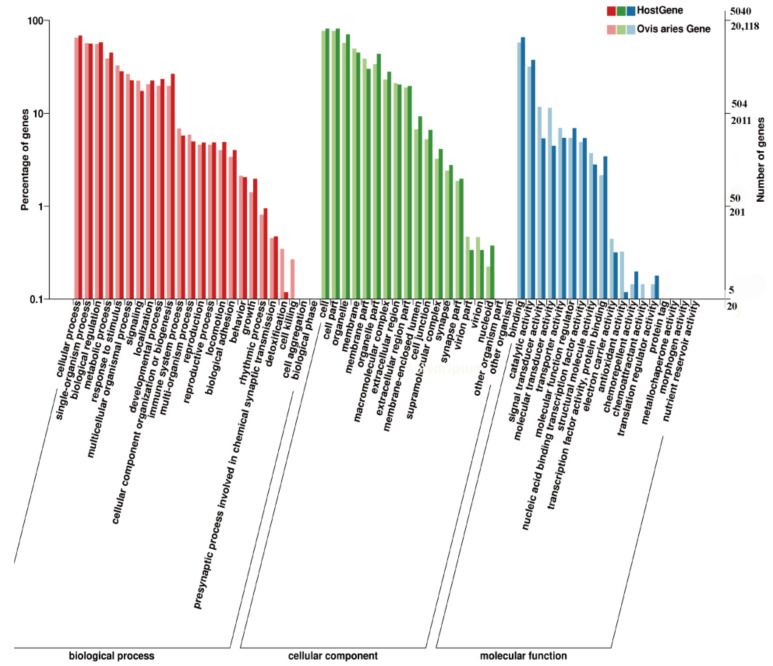
GO enrichment analysis for source genes.

**Figure 5 ijms-23-10488-f005:**
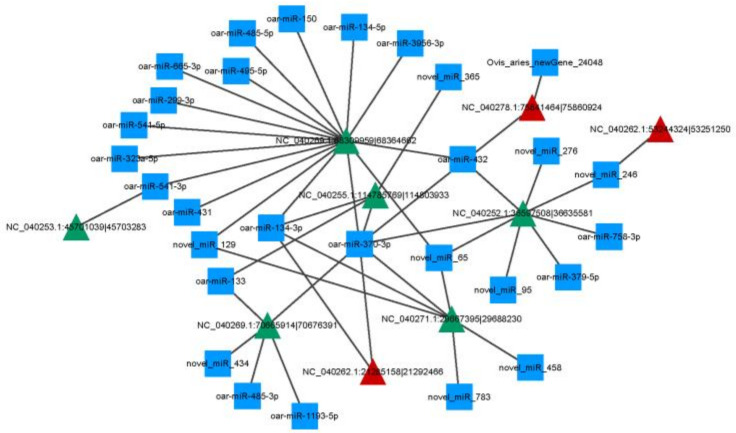
CircRNA- miRNA network. Note: The square represents miRNA, the red triangle represents downregulated circRNAs in the cold season, and the green triangle represents upregulated circRNAs in the cold season.

**Figure 6 ijms-23-10488-f006:**
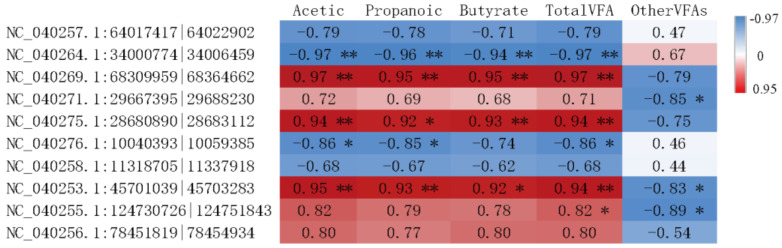
Pearson correlations between expression of circRNAs and VFAs. Note: Other VFAs refer to the sum of valeric acid, isobutyric acid, and isovaleric acid. ** refers to *p* < 0.01 and * refers to *p* < 0.05.

**Figure 7 ijms-23-10488-f007:**
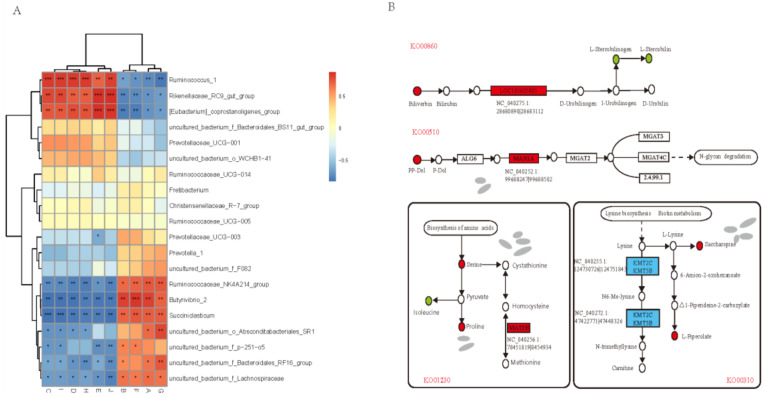
(**A**) Interaction network of circRNAs and microbiota; (**B**) interaction pathway of circRNAs and metabolites. NOTE: A indicated NC_040257.1:64017417|64022902, B indicated NC_040264.1:34000774|34006459, C indicated NC_040269.1:68309959|68364662, D indicated NC_040271.1:29667395|29688230, E indicated NC_040275.1:28680890|28683112, F indicated NC_040276.1:10040393|10059385, G indicated NC_040258.1:11318705|11337918, H indicated NC_040253.1:45701039|45703283, I indicated NC_040255.1:124730726|124751843, J indicated NC_040256.1:78451819|78454934.

**Figure 8 ijms-23-10488-f008:**
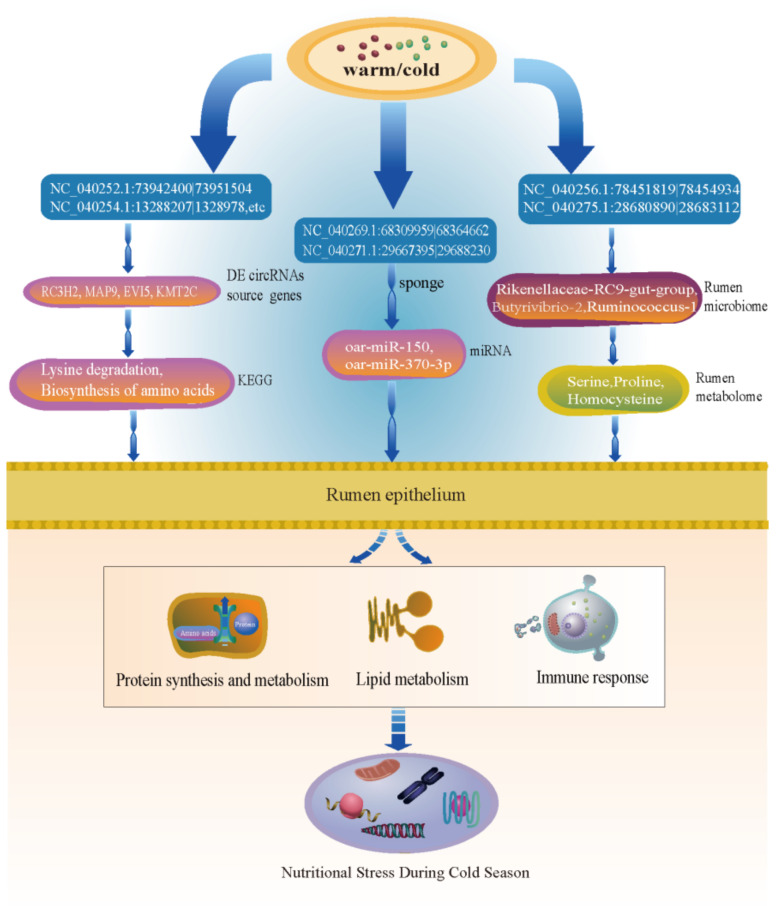
The regulation model of DE circRNAs in response to nutrient stress in the cold season.

**Table 1 ijms-23-10488-t001:** A list of primers used in the qRT-PCR.

CircRNA	Forward (5′→3′)	Reverse (5′→3′)
NC_040278.1:73454389|73479763	TAAGAACAGAAACACAAAATGCTC	CTTCCTCATCTCCAGGGTTT
NC_040255.1:124730726|124751843	AAGTTCACTCTGCATCAGG	ACTTCCCACAGAAAGGAC
NC_040274.1:49975759|49979215	CAGTGATGTGAGTGCGAGTA	GTGTCACTGGTGAAATCCTGT
NC_040259.1:83405933|83406542	ATGTGATCAGCGAGAAGCAG	CAGCATCGGTCTGTAACCTC
NC_040253.1:102980470|103003675	AAACGCAGGAACAGCTAGAG	CGCCACAGTCAGGTCTAATC
NC_040253.1:95968449|95969159	AGGTCAAAATTGAAAGTGAGGC	TCATTAATTTGTGAGGCTTGGAAT
β-actin	AGCCTTCCTTCCTGGGCATGGA	GGACAGCACCGTGTTGGCGTAGA

## Data Availability

The datasets presented in this study can be found in online repositories. The names of the repository/repositories and accession numbers can be found below: [Sequence Read Archive (SRA): SRR17883805-SRR17883810/SRR12719079-SRR12719088].

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
