# Peer review of "Coevolution of Rumen Epithelial circRNAs with Their Microbiota and Metabolites in Response to Cold-Season Nutritional Stress in Tibetan Sheep"

_ijms, 2022, doi:10.3390/ijms231810488_

Round 1

Reviewer 1 Report

The  authors conducted a good research on Co-evolution of Rumen Epithelial circRNAs with their Micro-2 biota and Metabolites to Respond to Cold-season Nutritional 3 Stress in Tibetan sheep.

However it needs improvement before publication, i have mentioned comments below 

L 11 which breed of sheep used in current research please add the name of the sheep breed. In abstract section also add the no of sheep used and duration of the trial.

It is better to add some line of materials and methods in abstract.

revised the conclusion section in Abstract part.

In L 35 withered ? 

In material method section please add Animal Ethical code of the university/institute

please rewrite line from Lines 82-89.

add the brief methods of the VFAs and Microbiota L 96.

Add few references in section 2.3 

L111-117 plz rewrite it and add references.

please add the primers accession No in list.

I suggest please check the methods of all parameters methodology and add references in each method.

why you have not use the Metaboanalyst for differencial metabolites and other results?

Fig.2 B is not clear please add the clear image.

Discussion section must be improved and validate the results with the previous studies.

please check the figure 8 carefully.

conclusion must be improved according to the  findings.

Author Response

Response to Reviewer 1 Comments

The authors conducted a good research on Co-evolution of Rumen Epithelial circRNAs with their Microbiota and Metabolites to Respond to Cold-season Nutritional Stress in Tibetan sheep. However it needs improvement before publication, i have mentioned comments below.

Point 1: L 11 which breed of sheep used in current research please add the name of the sheep breed. In abstract section also add the no of sheep used and duration of the trial.

Response 1: Thanks for your thoughtful suggestions on this paper. We added the name of the sheep breed, the no of sheep used and duration of the trial in abstract section.

Point 2: It is better to add some line of materials and methods in abstract.

Response 2: Thanks for your thoughtful suggestions on this paper. We have added some materials and methods in abstract. In this study, RNA-seq technology was used to identify differentially expressed (DE) circRNAs in the rumen epithelium of Tibetan sheep in cold and warm seasons, and to analyze interactions with the rumen microbiota and metabolites.

Point 3: Revised the conclusion section in Abstract part.

Response 3: Thanks for your thoughtful suggestions on this paper. We have made some modifications to the conclusion section in abstract part.

Point 4: In L 35 withered ?

Response 4: Thanks for your thoughtful suggestions on this paper. Withering grass stage (October-April of the following year) refers to the cold season. The withering of grass is a gradual process, here I should change withered to withering in line 53.

Point 5: In material method section please add Animal Ethical code of the university/institute

Response 5: We do thank you for your kind suggestion. We have added Animal Ethical code of the university/institute in material method.

Point 6: Please rewrite line from Lines 82-89.

Response 6: We do thank you for your kind suggestion. The irregular writing in lines 121-128 has been corrected.

Point 7: Add the brief methods of the VFAs and Microbiota L 96.

Response 7: We do thank you for your kind suggestion. We have added the brief methods of the VFAs and Microbiota.

Point 8: Add few references in section 2.3

Response 8: We do thank you for your kind suggestion. This part refers to the methods of RNA isolation, library preparation and circRNA sequencing, which were mainly determined according to the manufacturer’s instructions at BioMarker Technologies (Beijing, China). We have added references in section 2.4.

Point 9: L111-117 plz rewrite it and add references.

Response 9: Thanks for your thoughtful suggestions on this paper. We have rewritten it and added references in line 172-174.

Point 10: Please add the primers accession No in list.

Response 10: We do thank you for your kind suggestion. CircRNAs in this study were all novel circRNAs, so there was no accession number in NCBI.

Point 11: I suggest please check the methods of all parameters methodology and add references in each method.

Response 11: Thanks for your thoughtful suggestions on this paper. All parameters in methods have been measured and the relevant references have been cited.

Point 12: Why you have not use the Metaboanalyst for differencial metabolites and other results?

Response 12: Thanks for your thoughtful suggestions on this paper. Because these results have been published in other articles, they can only be cited.

Point 13: Fig.2 B is not clear please add the clear image.

Response 13: Thanks for your thoughtful suggestions on this paper. We have added the clear image in Figure 2 B.

Point 14: Discussion section must be improved and validate the results with the previous studies.

Response 14: Thanks for your thoughtful suggestions on this paper. We have made changes to the discussion. We have rephrased the sentences in the discussion and cited references.

Point 15: Please check the figure 8 carefully.

Response 15: Thanks for your thoughtful suggestions on this paper. We have modified Figure 8 and change sourse genes to source genes.

Point 16: Conclusion must be improved according to the findings.

Response 16: Thanks for your thoughtful suggestions on this paper. We have revised the conclusion.

Reviewer 2 Report

In the manuscript titled “Co-evolution of Rumen Epithelial circRNAs with their Micro-2 biota and Metabolites to Respond to Cold-season Nutritional 3 Stress in Tibetan sheep,” the authors explored host-microbe co-evolution by analyzing the interaction of rumen epithelial circRNAs and rumen microbiome and their metabolites. The author found strong correlations with circRNAs and volatile fatty acids under cold response. They also found correlations with circRNAs and amino acid biosynthesis pathways as well as glycan degradation. The latter is consistent with previously published literature. The authors provide an in-depth discussion of their results and hypothesize that the aforementioned interactions increase glycoside hydrolase activity (through co-encoding) facilitating microbial colonization of the rumen in cold environments, with amino acid biosynthesis and VFA production pathways aiding adaptation.

Comments

11)      Can the authors provide a brief sentence at the beginning of the abstract (after the first sentence) introducing the topic before discussing the results? This will provide a better transition to the results.

22)      Line 12. Do the authors mean 56 differentially expressed (DE) circRNAs from rumen epithelium tissue?

33)       Line 57. Do the authors mean, has participated in many biological processes?

44)      Lines 90-91. Do the authors mean, and the contents that were in it?

55)      On figure 6, what does the double asterisk represent? Please indicate what it means on the figure.

66)       Lines 260-261. Can the authors indicate the source genes KMT2C and KMT5b on figure 7B?

77)      Lines 281-282. Do the authors mean calcium ion?

88)       In figure 8, the authors should change sourse genes to source genes.

99)       In the text, the authors write Figure.number (e.g.,  Figure.8), which is different from what is written on their figures (e.g., line 388, Figure 8).

Author Response

Response to Reviewer 1 Comments

In the manuscript titled “Co-evolution of Rumen Epithelial circRNAs with their Microbiota and Metabolites to Respond to Cold-season Nutritional Stress in Tibetan sheep,” the authors explored host-microbe co-evolution by analyzing the interaction of rumen epithelial circRNAs and rumen microbiome and their metabolites. The author found strong correlations with circRNAs and volatile fatty acids under cold response. They also found correlations with circRNAs and amino acid biosynthesis pathways as well as glycan degradation. The latter is consistent with previously published literature. The authors provide an in-depth discussion of their results and hypothesize that the aforementioned interactions increase glycoside hydrolase activity (through co-encoding) facilitating microbial colonization of the rumen in cold environments, with amino acid biosynthesis and VFA production pathways aiding adaptation.

Point 1: Can the authors provide a brief sentence at the beginning of the abstract (after the first sentence) introducing the topic before discussing the results? This will provide a better transition to the results.

Response 1: Thanks for your thoughtful suggestions on this paper. We provided a brief material and method at the beginning of the abstract, which will be a better transition to the results.

Point 2: Line 12. Do the authors mean 56 differentially expressed (DE) circRNAs from rumen epithelium tissue?

Response 2: Thanks for your thoughtful suggestions on this paper. By sequencing rumen epithelial samples collected, a total of 56 differentially expressed (DE) circRNAs were found.

Point 3: Line 57. Do the authors mean, has participated in many biological processes?

Response 3: Thanks for your thoughtful suggestions on this paper. The references cited here showed that circRNAs were involved in cell proliferation, differentiation and apoptosis in studies that had been found. It may be my expression wrong and have been revised in 87-90.

Point 4: Lines 90-91. Do the authors mean, and the contents that were in it?

Response 4: Thanks for your thoughtful suggestions on this paper. This may be inappropriate but has been revised in line 130-131.

Point 5: On figure 6, what does the double asterisk represent? Please indicate what it means on the figure.

Response 5: Thanks for your thoughtful suggestions on this paper. Sorry for not annotating in the manuscript. We have revised ** refers to P<0.01 and * refers to P<0.05.

Point 6: Lines 260-261. Can the authors indicate the source genes KMT2C and KMT5b on figure 7B?

Response 6: Thanks for your thoughtful suggestions on this paper. We have edited the image Figure 7.

Point 7: Lines 281-282. Do the authors mean calcium ion?

Response 7: Thanks for your thoughtful suggestions on this paper. This refers to the GO pathways enriched by source genes of DE circRNA. These pathways are mostly related to calcium ion channels, and the GO pathways are omitted here. We have been revised in full text.

Point 8: In figure 8, the authors should change sourse genes to source genes.

Response 8: Thanks for your thoughtful suggestions on this paper. It is my writing error that has been corrected in Figure 8.

Point 9: In the text, the authors write Figure.number (e.g., Figure.8), which is different from what is written on their figures (e.g., line 388, Figure 8).

Response 9: Thanks for your thoughtful suggestions on this paper. We have revised the writing of the figures in the full text.

Round 2

Reviewer 1 Report

Point 12: Why you have not use the Metaboanalyst for differencial metabolites and other results?

Response 12: Thanks for your thoughtful suggestions on this paper. Because these results have been published in other articles, they can only be cited.

i did not understand the meaning of this response, if this data is already published in other paper, that how you have submitted this published data in new manuscript. little confusing please answer this quarry.

Author Response

Point 1: Why you have not use the Metaboanalyst for differential metabolites and other results?

Response: Thank you very much for your comments, and sorry for not being able to fully reply to you before. Firstly, Metaboanalyst is an online analysis method [1]. The differential screening conditions used in our article are the same as the method for differential analysis of Metaboanalyst, and we can draw pictures by ourselves during our analysis. The following is the process of our data analysis: After normalizing the original peak area information with the total peak area, the follow-up analysis was performed. Principal component analysis and Spearman correlation analysis were used to judge the repeatability of the samples within group and the quality control samples. The identified compounds are searched for classification and pathway information in KEGG [2], HMDB [3] and lipidmaps [4] databases. According to the grouping information, calculate and compare the difference multiple, T test was used to calculate the difference significance p-value of each compound. The R language package was used to perform OPLS-DA modeling, and 200 times permutation tests were performed to verify the reliability of the model [5]. The VIP value of the model was calculated using multiple cross-validation. The method of combining the difference multiple, the P value and the VIP value of the OPLS-DA model was adopted to screen the differential metabolites. The screening criteria are FC>1, P-value<0.05 and VIP>1. The difference metabolites of KEGG pathway enrichment significance were calculated using hypergeometric distribution test [6, 7].

Secondly, we performed the interaction analysis between the differential metabolites and the differential circRNAs, and found the functional pathway map that the differential metabolites and the source genes of differential expression circRNAs were co-enriched in this manuscript. We have specifically added the analysis steps of correlation analysis in section 2.10.

[1] Chong J, Xia J. MetaboAnalystR: an R package for flexible and reproducible analysis of metabolomics data [J]. Bioinformatics, 2018, 34(24): 4313-4314.

[2] Kanehisa, M. and S. Goto, KEGG: kyoto encyclopedia of genes and genomes. Nucleic Acids Res, 2000. 28(1): p. 27-30.

[3] David S W, Yannick D F, Ana M, An C G, Kevin et al. HMDB 4.0: the human metabolome database for 2018.

[4] Eoin F, Manish S, Dawn C, Shankar S. LIPID MAPS online tools for lipid research.

[5] Thévenot E A, Roux A, Xu Y, et al. Analysis of the Human Adult Urinary Metabolome Variations with Age, Body Mass Index, and Gender by Implementing a Comprehensive Workflow for Univariate and OPLS Statistical Analyses. Journal of Proteome Research, 2015, 14(8):3322-35.

[6] Yu G, Wang L, Han Y, He Q (2012). Clusterprofiler: an R package for comparing biological themes among gene clusters. OMICS: A Journal of Integrative Biology, 16(5), 284-287.

[7] Chong J, Xia J. MetaboAnalystR: an R package for flexible and reproducible analysis of metabolomics data. Bioinformatics, 2018, 34(24): 4313-4314.
